The role of Internet of Things (IoT) technology in modern cultivation for the implementation of greenhouses

ur Rehman Attique 1
Lu Songfeng 1
Ashraf Muhammad Awais 2
Iqbal Muhammad Shahid 3
Nawabi Awais khan 4
http://orcid.org/0000-0002-6385-5511 Amin Farhan 5 farhanamin10@hotmail.com
Abbasi Rashid 6
de la Torre Isabel 7 isator@uva.es
Gracia Villar Santos 8
http://orcid.org/0000-0002-0800-8563 Lopez Luis Alonso Dzul 8
http://orcid.org/0000-0001-5307-9582 Heyat Md. Belal Bin 9
1 School of Cyber Science and Engineering, Huazhong University of Science and Engineering (HUST) , Wuhan, Hubei , China
2 School of Information Engineering, Chang’an University , Xi’an, Shaanxi Province , China
3 Department of Science and Information Technology, Women University of AJ&K , Bagh , Pakistan
4 School of Computer Science and Engineering, Central South University , Hunan, Changsha , China
5 School of Computer Science and Engineering, Yeungnam University , Gyeongsan , Republic of South Korea
6 College of Computer Science and Artificial Intelligence, Wenzhou University , Wenzhou , China
7 Department of Signal Theory and Communications, University of Valladolid , Valladolid , Spain
8 Isabel Torres, Universidad Europea Del Atlántico , Santander , Spain
9 School of Engineering, Westlake University , Hangzhou , China
Sajid Ullah Syed
Electronic publication date: 2024 Sep 26
Publication date: 2024
Volume: 10
Electronic Location ID: e2309
Received 2024 Apr 22; Accepted 2024 Aug 15
Copyright: © 2024 ur Rehman et al.
Copyright year: 2024
Copyright holder: ur Rehman et al.
License: This is an open access article distributed under the terms of the Creative Commons Attribution License, which permits unrestricted use, distribution, reproduction and adaptation in any medium and for any purpose provided that it is properly attributed. For attribution, the original author(s), title, publication source (PeerJ Computer Science) and either DOI or URL of the article must be cited.
License URL: https://creativecommons.org/licenses/by/4.0/

Keywords: Internet of things, Big data, Security, Technologies

Funding: European University of Atlantic This research is funded by the European University of Atlantic. The funders had no role in study design, data collection and analysis, decision to publish, or preparation of the manuscript.

==============================
In recent years, the Internet of Things (IoT) has become one of the most familiar names creating a benchmark and scaling new heights. IoT an indeed future of the communication that has transformed the objects (things) of the real world into smarter devices. With the advent of IoT technology, this decade is witnessing a transformation from traditional agriculture approaches to the most advanced ones. Limited research has been carried out in this direction. Thus, herein we present various technological aspects involved in IoT-based cultivation. The role and the key components of smart farming using IoT were examined, with a focus on network technologies, including layers, protocols, topologies, network architecture, etc. We also delve into the integration of relevant technologies such as cloud computing, big data analytics, and the integration of IoT-based cultivation. We explored various security issues in modern IoT cultivation and also emphasized the importance of safeguarding sensitive agricultural data. Additionally, a comprehensive list of applications based on sensors and mobile devices is provided, offering refined solutions for greenhouse management. The principles and regulations established by different countries for IoT-based cultivation systems are presented, demonstrating the global recognition of these technologies. Furthermore, a selection of successful use cases and real-world scenarios and applications were presented. Finally, the open research challenges and solutions in modern IoT-based cultivation were discussed.

Introduction

The idea of the Internet of Things (IoT) captured attention in 1999 using auto-id at MIT, which is a core center, and the investigation publications of its relevant market. All the attention has been caught by the concept of IoT. IoT is defined as the integration of various sensing devices that interact and exchange information about external and internal states over the internet. IoT is being considered as one of the megatrends for the technologies of next-generation (Lee & Lee, 2015). With its extended benefits of providing advanced connectivity to services, and end devices, the system is leaving a powerful impact on the business spectrum. Several appropriate solutions are being offered by IoT for different applications including smart cities, healthcare, greenhouses, etc. (Mora et al., 2024). In the field of cultivation, a significant amount of work has been done to develop IoT-based solutions for smart green housing. In green housing, multiple challenges and complications can be addressed through IoT by providing smart solutions (Ray, 2017). IoT has done an exceptional job in the cultivation environment and more research is expected on tracking the challenges, such as cost management, water and productivity shortage, etc. Mitigating these issues will provide ease to farmers because the issues are increasing and considered to be the most faced complications in green housing (Farrukh, Mathrani & Sajjad, 2024). There are already a few technologies provided by the art that only encountered the issues faced in green housing but also presented a quick fix to boost productivity by lowering the amount charged (Sadek & Shehata, 2024). All the work done on networks of wireless sensors has come out to be beneficial because it enables us to send the relevant data to the main servers which we collect through sensing devices (Hosny, El-Hady & Samy, 2024). The data that we collect through sensors not only gives information that we require about many various environmental conditions but also makes that information useful for us in terms of supervising the whole system more efficiently (Tenzin et al., 2017). Supervising the environmental conditions is not the only factor for the crop’s evaluation because multiple other factors have affected the productivity of the crops (Hosny, El-Hady & Samy, 2024; Jayaraman et al., 2016). The other factors included in greenhouse management are unwanted image movement, attack of dangerous animals and thefts, and monitoring of soil, and crop (Padalalu et al., 2017). IoT is doing every possible work either in terms of putting a restriction on resources or scheduling those resources that are restricted to make sure to enhance production (de Oliveira et al., 2017).

The agricultural sector is undergoing a technological revolution with the adoption of IoT technology, particularly in greenhouse management. IoT enables precise and automated monitoring and control of environmental conditions, providing a reliable, predictable, and efficient approach to modern cultivation. This summary provides a comprehensive overview of the state of the art in IoT-based greenhouse management, focusing on network technologies, data analytics, security, and regulatory frameworks (Vinson Joshua et al., 2022).

Figure 1 shows trends of cultivation providing accessible and amount effective cooperation through guarded and undamaged connectivity across individual greenhouses through machine learning techniques (Tenzin et al., 2017; Vinson Joshua et al., 2022). Whereby, wireless devices are normally used in its cultivation networks to enable the monitoring of crops in real-time. The liberalism which is a smart IoT vertical kit in cultivation, and the extreme sensor kit for plant monitoring are the two sensor kits that have been deployed to monitor the moisture of clay, humidity, the wetness of leaf, temperature, productivity, and flow of air (Marimuthu et al., 2017). The servers, gateways, and databases of cultivation are playing a great role in storing the records of cultivation and providing authorized users the services on demand (Shirsath et al., 2017). As a whole, there are many different applications, protocols, and prototypes in the cultivation field.

Figure 1 The recent trends in cultivation.

Designed by Freepik (https://www.freepik.com/).

This research work provides a comprehensive and distinctive overview of the role of IoT technology in modern greenhouse cultivation. By integrating aspects of network architecture, data analytics, security protocols, and regulatory frameworks, it offers a broader perspective than previous research. The success stories and differentiation factors make this a valuable resource for researchers and practitioners aiming to implement IoT in greenhouse management effectively.

The agricultural sector is facing unprecedented challenges due to rapid population growth, climate change, and increasing demand for sustainable farming practices. Traditional cultivation methods often lead to inefficiencies, resource wastage, and suboptimal crop yields, particularly in greenhouse environments where precise control over climatic conditions is essential for optimal production. Greenhouses offer a controlled environment that can significantly improve crop yields and resource efficiency, but managing these environments effectively requires real-time monitoring and automated decision-making, which conventional methods cannot adequately provide.

Key challenges in greenhouse cultivation

Resource optimization: Traditional greenhouse management often leads to excessive use of water, fertilizers, and pesticides due to a lack of accurate, real-time data.

Monitoring and control: Conventional methods rely heavily on manual monitoring, which is time-consuming and prone to human error, leading to suboptimal environmental control.

Lack of network infrastructure: The integration of IoT technology is hampered by fragmented network protocols, limited bandwidth, and unreliable communication between devices.

Data analytics and insights: While IoT sensors can generate large volumes of data, there is a lack of effective data analytics tools to translate this data into actionable insights for better decision-making.

Security and privacy concerns: IoT systems in greenhouses are vulnerable to cyber threats, posing risks to data integrity, privacy, and the overall reliability of the system.

Regulatory compliance: Implementing IoT-based greenhouse systems across different regions is complicated by varying standards and principles, leading to regulatory compliance issues.

Cost and accessibility: The high cost of IoT infrastructure and technology makes it challenging for small-scale farmers to adopt IoT-based solutions, limiting the widespread application of smart farming practices.

Research GAP

Despite significant advancements in Internet of Things (IoT) technology for greenhouse cultivation, several gaps hinder the effective implementation and optimization of IoT-based greenhouse management systems. Comprehensive network architecture frameworks: Existing studies mainly focus on isolated aspects of network architecture, such as protocols and topologies. A comprehensive framework covering all layers and specifically tailored for greenhouse environments is missing.

Integration of cloud computing, Big Data, and Edge Analytics: While the importance of cloud computing and big data analytics is recognized, their seamless integration with edge computing remains underexplored. Efficient data storage, processing, and real-time decision-making require further research.

Security protocols and privacy mechanisms: Although security concerns are often highlighted, there is a lack of comprehensive, multi-layered security frameworks tailored specifically for IoT-based greenhouse environments.

Standardization and interoperability: Fragmentation due to varying protocols, devices, and platforms leads to interoperability issues. Universal standards and protocols are needed for seamless integration between IoT devices.

Regulatory compliance and global standards: Research has focused on technological aspects, with limited attention to regulatory frameworks. A comprehensive analysis of global standards and principles is required for international adoption.

Cost-effectiveness and accessibility: IoT technology remains expensive, limiting adoption among small-scale farmers. Research is needed to develop scalable and affordable IoT solutions for widespread use.

Application-specific IoT solutions: Most IoT solutions are generic and not tailored to specific crops or environmental conditions. Crop-specific and region-specific solutions are needed for optimizing resource usage and yields.

Research objectives

To analyze the role of IoT technology in modern greenhouse cultivation, including network architecture, protocols, and data analytics.

To identify and address the challenges related to network infrastructure, data management, security, and regulatory compliance in implementing IoT-based greenhouse systems.

To present success stories and real-world applications that showcase the benefits of IoT in greenhouse management.

To establish a clear framework for the adoption of IoT technology in greenhouse cultivation, providing guidelines for sustainable and efficient agricultural practices.

IoT research trends related to cultivation include platform as well as the architecture of the network, applications, security, and challenges among others. Moreover, different policies of IoT and guidelines have been deployed in the relevant fields over the globe in many organizations and countries (Wang, 2024). However, a reasonable amount of work has been done in the IoT cultivation environment and elaborated research on IoT in the cultivation frame of reference to figure out the present research state is needed for transforming the technologies of cultivation through the innovation of IoT. This article manages to analyze the various problems and ongoing movements in IoT smart green housing (Patil et al., 2017).

Motivation of our research

The IoT and green housing a popular areas but limited research has been carried out in this direction. Thus, the basic motivation of this research is to identify the proposed solutions for greenhouse farming by monitoring, tracking, predicting, and controlling. The key contributions are given below. In this research, we presented an in depth review of various IoT based applications, network, topology, architecture, platforms used for the green housing.

We explored main factors and applicable technologies for IoT-established smart green housing.

We presented various applications, use cases and the applications used for the green housing.

We proposed an IoT-based greenhouse farm management taxonomy and attacks taxonomy. By addressing key challenges and providing a clear framework for the implementation of IoT technology in greenhouses, this research contributes valuable insights for researchers, practitioners, and policymakers working towards sustainable and efficient agricultural practices.

We have presented various challenges for instance; networking challenges and hardware, software and technical etc. and future directions in this area to overcome the current gaps.

The rest of the article is organized as follows. In “Search Methodology” we discuss the main factors and the applicable technologies for IoT technologies. In “IoT Technology and Greenhouse”, we discuss IoT technology and greenhouses. In “IoT Cultivation Architecture”, we discuss IoT cultivation architecture. In “The role of Internet of Things (IoT) Technology in Modern Cultivation”, we discuss the role of IoT technology in modern cultivation. In “IoT and Greenhouse Security” we discuss IoT and green security, in “Smart Management Systems for Greenhouses” we discuss the challenges in this area. Finally, “Key Challenges and Research Directions in Green Houses and Modern Cultivation” discusses the conclusion of this study.

Search methodology

In this section, we present the search methodology. To ensure comprehensive and unbiased coverage of the literature on smart green housing using IoT, a multi-faceted approach is crucial. Here’s a detailed explanation of how this can be achieved:

Database and journal search

Utilize multiple academic databases such as Google Scholar, IEEE Xplore, and ScienceDirect to gather a broad range of research articles and conference articles. Include specialized journals in the fields of sustainable architecture, environmental engineering, and information technology.

Keyword strategy

Develop a comprehensive list of keywords and phrases such as “smart homes”, “green buildings”, “IoT in housing”, “sustainable smart architecture”, and “energy-efficient IoT solutions”. Use Boolean operators to expand or narrow the search results (e.g., “IoT AND ‘green housing” or “smart homes‘ OR ’sustainable living”).

Inclusion of grey literature

Incorporate grey literature such as technical reports, government publications, and white articles to ensure coverage of non-commercial, cutting-edge innovations and regulations in the field.

Citation analysis

Perform forward and backward citation tracking on pivotal studies to discover seminal work and the latest research developments.

Cross-disciplinary focus

Encourage a cross-disciplinary approach by reviewing literature that intersects with other fields such as urban planning, renewable energy technologies, and behavioral sciences.

International coverage

Include studies and reports from various geographical locations to understand regional advancements and challenges in implementing IoT in green housing.

Review articles and meta-analyses

Study review articles and meta-analyses for summaries of existing research and gaps in the literature, providing insights into the evolution of the field.

Consultation with experts

Engage with experts in smart housing and IoT through interviews or consultations to gain insights on emerging trends and critical issues not yet fully addressed in academic literature. By following these strategies, a comprehensive and balanced understanding of the current state and future directions of smart green housing using IoT can be achieved, minimizing bias and maximizing the breadth and depth of the gathered information.

Main factors and applicable technologies for IoT-established smart green housing

The technologies of IoT based on smart green housing with its major factors and components are presented in this section.

Main factors of IoT established smart green housing

IoT-based smart green housing is made up of four major factors as shown in Fig. 2. Multiple tasks are performed by a sensor like sensing clay, temperature, weather, light, and moisture. Similarly, many control functions are performed by devices like the discovery of nodes, identification of devices and services getting names, etc. All mentioned work or operations are achieved by any sensor or device that is being regulated by a microcontroller. Any remote device that performs such controlling operations is linked via the internet. Acquisition of data is divided into two subunits namely, the Internet of Things and standard acquisition of data. The IoT acquisition of data components consists of seven protocols that are MQTT, AMQP, node, COAP, DDS, and HTTP. For the implementation of smart green housing, more protocols can be used depending upon the requirements and condition, although, in the standard (principle) acquisition of data, ZigBee, Lora-wan, Wi-Fi, isobars, and SigFox protocols have been used (Shirsath et al., 2017; Wang, 2024). Processing of data composed of various features including the processing of video and image, data burdening along with mining and, supported decision-based system as shown in Fig 2. Under the needs of the system, there is a chance of adding a new feature that may function in parallel to support other assistance or services.

Figure 2 The main components of IoT established greenhouse.

Designed by Freepik (https://www.freepik.com/).

Analysis of data consists of mainly two features which there is monitoring and controlling (Patil et al., 2017). The main application of smart cultivation is the monitoring of the greenhouse. The Internet of Things enables the monitoring of plant applications to report various types of field conditions like the richness of soil, humidity, temperature, pressure, and gas (Locatelli et al., 2024). The smart design of the greenhouse eliminates the intervention that is manual and measures different parameters of climate by super-intelligent devices and sensors of the Internet of Things according to the requirements of plants (Locatelli et al., 2024).

Relevant technologies of green housing

In recent years, there have been some important improvements in the area of artificial technology and smart technologies regarding cultivation and smart greenhouse management systems, we have a huge variety of such technologies that can’t be implemented at once. Research demonstrates the successful deployment of a wireless sensor network and IoT platform for intelligent animal monitoring. By integrating real-time data analytics, wearable sensors, and wireless communication, the system provides a scalable and sustainable solution for modern livestock management. The comprehensive insights into animal health and behavior can significantly improve farming practices, ultimately contributing to more sustainable and productive livestock farming (Arshad et al., 2022). For the smart cultivation and greenhouse system, we will focus on a few vital technologies. Therefore, in the following we have discussed the important technologies here:

Networks and technologies

Various kinds of ranges either in terms of short or long are made up of an IoT cultivation network for interaction. Many Internet of things-based technologies of network support in designing plant-supervising sensors and plant-supervising devices. In the system and applications of IoT, cultivation network interaction protocols are considered to be the backbone (Priya et al., 2024). Over the network, they help in interchanging the information or data.

Cloud fog and edge computing

Cloud association with the Internet of Things and network over this provide access to the cultivation resources over the network for remote access. The main purpose of this network over the cloud system is to make various cultivation needs to perform a specific task to do some vital functions (Al-Sarawi et al., 2017).

For this technology, a new cloud-based software infrastructure has been introduced to make data more accurate and information-friendly. For the perfect data collection, IoT edge computing is the next hot topic for cloud and network data communication to facilitate the next upcoming generations. The data from actuators and sensors and other embedded devices have the data over the cloud (Locatelli et al., 2024; Al-Sarawi et al., 2017; Zamora-Izquierdo et al., 2019). This technology confirms the relativity of requirements and features to do work simultaneously, fog and edge computing are the hot and important technologies to have in our project (Kumar & Patel, 2014).

Robotics

For smart green housing, multiple robots of robotics have progressed to help in minimizing the number of farmers in number and it is done by keeping the speed high in work through new techniques (Kumar & Patel, 2014). Robots execute elementary functions such as weeding, spraying implanting, etc. All mentioned robots are regulated by the use of IoT to help in increasing the production of crops and the utilization of powerful resources. A robotics-based multi-sensor approach has been suggested for ground mapping and characterization (Pavón-Pulido et al., 2017).

Machine learning and big data analytics

The next new technology is data science to be used in our project and consists of deep and machine learning to handle a large amount of data to use for the essential functional analytics on data produced by greenhouse sensors (Milella, Reina & Nielsen, 2019). These analytics give different important crop analyses in different areas. A good and better semantic review can give our data a presentable sense. Neural networks for optimal results to have a project on high speed. Neural networks are quite recognized because optimal solutions are supported at high speed by them. Hacker and outsider attacks are detected by this advancement, again in this neural network, one more additional factor that can be added is the model and training of data through using IoT systems. Module and data training by using IoT systems (Pavón-Pulido et al., 2017).

Iot technology and greenhouse

The IoT network for cultivation is the main basic part of IoT in cultivation. It sends and receives the cultivated data and it also tracks the cultivated data. Figure 3A shows the IoT-cultivated network platforms, IoT-cultivated network topologies, and IoT-cultivated architecture and the protocol used in the cultivation network.

Figure 3 (A) IoT greenhouse network; (B) the layer structure of the 6lowpan.

Designed by Freepik (https://www.freepik.com/).

The authors highlight the effectiveness of a GSM-based solar-powered automatic irrigation system using environmental sensors. The combination of renewable energy, sensor data, and mobile communication provides a sustainable and efficient solution for modern irrigation challenges. The system significantly improves water usage efficiency and offers a scalable solution for farmers seeking to optimize irrigation practices (Rehman et al., 2017).

IoT greenhouse network architecture

As we are using IoT in the cultivated domain, the IoT cultivated network is an essential part of this field. The IoT cultivated network architecture in Fig. 3A proposes the IoT-cultivated network’s physical portion and specifies the principles and techniques that explain all the elements of a physical network (Milella, Reina & Nielsen, 2019). Applications of IoT usually use the most popular four layers of architecture or protocol and they can exchange data for useful processing by using IP (Mehra et al., 2018).

The current state of work on IoT and greenhouse technology highlights the transformative potential of integrating advanced technologies into greenhouse management. By addressing issues such as interoperability, security, and cost-effectiveness, IoT-enabled greenhouses are paving the way for innovative, sustainable, and efficient agricultural practices. However, challenges remain, particularly in the areas of standardization, security, and global adoption, which future research aims to address (Priya et al., 2024).

After surveying different layers of protocol, we get two more approaches that are 6lowpan and ipv6 as shown in Fig. 3B. These approaches are the terminating level of hardware/software abstraction development that can use many applications from different users (Vistro et al., 2020). It utilizes communication protocols and monitors different parameters in cultivation like irrigation monitoring, soil moisture values, weather information, etc. (Jan & Toldan, 2023).

Iot cultivation architecture

The cloud model and big data analytics model both deal as a platform with an IoT cultivation network platform. The latest study presents an intelligent greenhouse monitoring and control system combining sensors, embedded systems, and an IoT platform. By automating environmental control, the system enhances resource efficiency and optimizes crop yields, making it a valuable solution for modern greenhouse cultivation. The successful field deployment and evaluation highlight the system's potential to significantly improve greenhouse management practices, contributing to sustainable agriculture (Arshad et al., 2020).

Big data components

By combining the analytics of big data with the Internet of Things the world becomes smart. The physical thing is linked to the internet and easily can access the data of sensors and control the smart world from a distance by using IoT devices. It checks out the data sets of large amounts and makes conclusions. Big data analyzers are used when there is a large amount of data and they detect meaningful information according to the requirements in different formats of data. The greenhouse disease control and the growth of plants model are made on the premises of greenhouse data. Optimal cost analysis and plant productivity services are provided by big data which is mentioned in Fig. 4.

Figure 4 Big data analysis-based IoT greenhouse network platform.

Designed by Freepik (https://www.freepik.com/).

IoT cultivated network platform is shown in Fig. 4 based on big data analytics. The network platform model consists of six components: (1) user experience, (2) monitoring and sensing, (3) big data analytics, (4) storage services, (5) physical implementation, and (6) communication protocol. Big data analyzer is the backbone of IoT that helps to collect a large amount of information like miniaturization, weather conditioning, online crop monitoring, soil fertility, etc.

Predictive analysis

IoT technology and smart green housing market intelligence combination make the greenhouse environment smart by using predictive analysis. Predictive analysis’s main responsibility is to analyze the cultivation information then explore the digital awareness of data and process the information. For checking the probability of plant production for the next season predictive analysis is made. We use different sensors to check the weather conditions, plant disease, and plant production, then calculate profit/loss. Predictive analysis helps the manager of the greenhouse to understand the different techniques and the perfect time for planting.

Multicultural analysis

The multicultural analysis defines multiple forms of cultivation. The multicultural analysis allows big data analyzers to reduce the risk of plant demolition in different ways. The botanic aquaculture layer is merging with big data for the enhancement of the growth rate of water. When we allow big data to enable multicultural techniques such as horticulture, floriculture, and citriculture then they utilize many benefits. It is very helpful for greenhouse managers because it makes decisions related to pest control and plant seasonal growth. When it comes to the cultivation of earthworms, vermiculture comes into use. Arboriculture is used since it is considered to be useful for the cultivation of woody plants. An app of olericulture is used for several vegetables in terms of measuring their growth rate and prediction as well.

Storage services

For good analysis, the greenhouse manager stores information for future use and is also used for the best and more productivity in different seasons. The layered framework is shown in Fig. 5.

Figure 5 Greenhouse information service model functional framework.

Designed by Freepik (https://www.freepik.com/).

Communication protocols

Communication channels gather plant and crop data from the local controller. As data is collected through communication protocol and then they encapsulate the data. Communication channels provide the communication between the sensor/device and microcontroller using the i2c protocol. The nerve center transmits and processes the data of IoT in cultivation by using the above-mentioned protocols. It consists of technologies related to the internet such as Lora Wan, CDMA (code division multiple access), and Wi-Fi technology. For long-distance communication, we use ZigBee as an enabler when a third party is not available for services like long-term evolution (LTE) and CDMA or global system for mobile (GSM). ZigBee technology is used to gather data accessible by a web server available in the gateway.

Physical implementations

For monitoring multiple cultivated applications, we use different sensors, microcontrollers, and actuators that are physically implemented. Routers, gateways, and switches are implemented in the physical layer as network equipment. The microcontroller’s main function is to gather data by multiple sensors connected to it and connect to the next layer. A microcontroller can be self-powered using solar panels, battery power, and self-power with backup batteries depending on the application. For the microcontroller, we use a Raspberry Pi2 single board with 3.7v lithium battery power in every front-end node. Cultivated data is sensed according to environmental conditions and then predefined instruction is used for actions using an actuator. The manager role is played by a microcontroller for networking operations and other operations are performed by sensors and actuators. Frameworks of IoT functions are shown in Fig. 5 it explains how managers and cultivators can access the different databases from the application layer as a support layer. All operations are stored in the business layer which is essential for IoT cultivation. The session layer is connected with the data acquisition layer through some IoT-based protocols such as AMQP, COAP, MQTT, etc. The acquisition layer contains the processing layer, perception layer, and transmission layer. The perception layer mostly distributes sensors in the greenhouse at terminal nodes. Data is collected through terminal nodes and sent to the sink node which transmits data to the transmission node in the transmission layer. Through GPS and network, the data is sent to the main server after processing data in the processing layer. Some IoT communication technologies use cps nodes to connect actuators and sensors. The edge layer arranges the greenhouse main control and the edge layer is made up of four modules of control for nutrition, climate, irrigation, and auxiliary tasks.

Greenhouse manager experience

To monitor the plant’s productivity, it has been used to design the greenhouse manager experience layer. Managers can monitor the production in different ways e.g. Appropriate fertile selection is identified for best growth and productivity of plants. Plant growth conditions, climate conditions, and soil quality monitoring are very helpful for the managers and they can easily migrate the less productivity problem. Big data analytics and IoT also improve the quality of plants/crops produced in greenhouses and it is done by control measures and ecological forecasting. Big data recommend different types of pesticides to greenhouse managers for the best quality of the product. For precise cultivation, big data analysis also provides the perfect amount of fertilizer for the soil nutrient.

Cloud infrastructure

As we know cloud computing provides us storage to store a large amount of data and it is connected to a virtualization server and performs some actions accordingly. For precise cultivation, IoT base cloud design is presented. Sensors manage and analyze the cultivation data which use IoT techniques and methods. It generates information through sensor devices for making an accurate decision (Arshad et al., 2020). Cloud-based IoT cultivated network design is recommended as shown in Fig. 6.

Figure 6 Cloud-based IoT green house network platform.

Designed by Freepik (https://www.freepik.com/).

Four layers are proposed based on platforms such as gateway, hardware module, cloud storage, and fog computing. Cultivation-related data like soil, crop, fertilization, weather, and cultivation marketing is centralized in the cloud storage layer and provides resources on user demand through the network infrastructure. Internet or cloud installs web-based services and analysis resources accessible by cloud services. The major aim of sensors and devices is sensing the data but it cannot share the data only by connecting to the internet. Local gateway is designed to overcome the data sharing problem but the gateway is working as a bridge between sensors security, controllability, connectivity, and hardware devices. When novel sensors and actuators are automated it works as a gateway. Greenhouse implements gateways that improve the working in real-time to control the automation monitoring system in the greenhouse. Fog computing integrates resources and distributes cloud services and hardware modules. Fog computing ensures the processing in real-time and reduces the computational load.

We proposed the basic purpose of fog computing in the platform of networking is to provide the scalability of cloud computing on user demand and take advantage of both edge and cloud computing. The central processing unit, actuators, microcontrollers, and sensors are implemented for monitoring as a hardware module and sensing the cultivation variables. The hardware module is used to create the process and services and distribute it in local or global networks. Fast response time is implemented in a smart greenhouse and has data exchanging capability when it is needed. MQTT (message queuing telemetry transport) and rest (representational state transfer) both have the capabilities to exchange data in a fast response time. A distributed system is more effective as compared to big data for smart greenhouses. The distributed system breaks the large computational problem into a smaller and easier task such as nutrients, climate, temperature, soil moisture, crop, etc.

Network topology and protocols of IoT greenhouse

The topology network of IoT cultivation presents the sequence of numerous fundaments of the Internet of Things cultivation network. The unique or best scenario for smart green housing is also represented by this. Figure 7 shows how the relevant data has been collected from multiple sensors by grid computing. The sensors include moisture, temperature, gas, smoke, humidity, ultraviolet, etc. by using multiple sensors, grid computing successfully constructs the network for cultivation topology.

Figure 7 Ideal illustration of IoT-based present greenhouse solutions.

Designed by Freepik (https://www.freepik.com/).

The cultivation solution which is ubiquitous supports transforming the capacity of storage for numerous portable electronic devices including portable computers, and terminals into the grids of hybrid computing and smartphones. Figure 8A visualizes a scenario where all plant parameters are supervised by deploying cultivation devices and other sensors all over the area. Later on, the sensing data that is collected through devices and sensors is stored and this process becomes quite successful for cultivation. The greenhouse managers supervise the plant variables specifically from anywhere depending upon the aggregation and analysis. For the streaming of cultivation videos, a properly configured network is provided by topology. Figure 8A shows through an interconnected network with different protocols of the internet and accessing the gateway network it’s easy to support the streaming of pests. The protocols include GSM and WiMAX.

Figure 8 (A) Greenhouse remote monitoring greenhouse (B) low power WSN topology.

Designed by Freepik (https://www.freepik.com/).

Figure 8A shows the different scenes for which any greenhouse farm can be monitored by different smart devices spread all over the field. All the devices are useful when they store and gather multiple data at a time, to be aggregated. This will allow the greenhouse farm manager to have a keen eye on the multiple variables of the field remotely. This includes the network topologies and proper network configuration for having multiple data. Figure 8A has the GSM module to have direct data of pests using the internet protocol and WIMAX as a gateway.

WSN topology of low power

Figure 8B shows a topology where a sensor network is wireless has low power has been constructed to supervise and regulate the different factors of cultivation. ZigBee is a topology which has that has been used for the transmission of data on the area of cultivation. Different sensors are found at end devices. The sensors include UV, PIR, motion, soil, humidity, gas, and a microcontroller.

The controller and router give direct connection to end devices. During a connection, the controller uses a serial port to interact with the base station to analyze the received data or information. If we look at the perspective of software monitoring each device that is at the end is perfectly started and connected sensors are turned on appropriately. The moment sensors get activated, all devices that are connected follow the router to connect the way they were constructed. After having confirmation all devices that are at the end use the identical key to have a possible connection with WSN. Data is collected through sensors and sent collected data to the base station to get analyzed at the receiving end. The sensors are being read which are attached with devices at the end and after being read they get ready for transmission through ZigBee to the router. This net-based topology has the major benefit of interacting bi-directionally through bringing ZigBee into use.

Protocols for greenhouse in IoT

There are several protocols for the communication of IoT. These IoT protocols are being used widely for the means of making green housing smart enough. Bringing these protocols in use will help the greenhouse managers communicate in a way that’s convenient enough. The greenhouse managers will also be able to enhance the growth of plants by efficient decision-making in smart green housing. The wireless common protocols that are being utilized include ‘WiMAX’, ‘LORAWAN, ‘RFID, ‘low-rate wireless personal area networks’, ‘2g/3g/4g-mobile interaction protocol, IEEE 802.11 Wi-Fi, Bluetooth, ZigBee, and contrast wireless protocols are listed in Table 1.

Table 1 Comparison of existing wireless protocols.

Parameters	Energy consumption	Transmission range	Cost	Frequency band	Data rate	Standard	
Wimax	Medium	<50 Km	High	2–66 GHz	1 Mb/s–1 Gb/s (Fixed) 50–100 Mb/s (mobile)	IEEE 802.16	
ZigBee	Low	10–20 m	Low	2.4 GHz	20–250 Kb/s	IEEE 802.15.4	
Wi-Fi	High	20–100 m	High	5–60 GHz	1 Mb/s–7 Gb/s	IEEE 802.11 a/c/b/d/g/n	
MQTT	Low	–	Low	2.4 GHz	250 kbps	OASIS	
RFID	Low	1–5 m	Low	860,960 MHz	40 to 160 kbit/s	ISO 18000-6C	
Sigfox	Low	30–50 km	Low	200 kHz	100–600 bit/s	SigFox	
Mobile communication	Medium	Entire cellular area	Medium	865 MHz, 2.4 GHz	2G: 50–100 kb/s 3G:200 kb/s 4G: 0.1–1 Gb/s	2G-GSM, CDMA.3GUMTS, CDMA2000, 4G-LTE	
Bluetooth	Low	1–24 Mb/s	Low	24 GHz	1–24 Mb/s	IEEE 802.15.1	
LR-WPAN	Low	40–250 Kb/s	Low	868/915 MHz, 2.4 GHz	40–250 Kb/s	IEEE 802.16	
LoRaWAN	Very low	<30 KM	High	868/900 MHz	0.3–50 Kb/s	LoRaWAN R1.0	

The role of internet of things (iot) technology in modern cultivation

In the past recent years, the smart management system has been a hot topic for discussion of the people of different areas and fields so is for the farms. Farms like poultry farms, vegetable farms, and other greenhouse and horticultural farms like greenhouse are turned into well-furnished managed smart farms by using systems like GSM and Wi-Fi. A farmer can easily access those things which he uses to manage manually. The author introduces the management system called the farm management system to overtake problems like manual operations etc. The farmer gets all the data processed to use for the betterment of the farm and easily increases the production rate. For this purpose, they use sensors and actuators for the collection of data, and for further use, the data is processed by the smartphone, directly by computer, or by application via the internet. This will optimize the water usage; fertilizer usage and weather conditions can be used according to need. Apart from this actuators can also call for action against pests and other intruders.

IoT application in greenhouse management

For making the effective small green housing productivity more enhancing and beneficial, more efficient resources are created by using the numerous applications of IoT effective small cultivation which is shown in Fig. 9. The domains of applications of IoT effective small cultivation which are mainly used are farm animals monitoring, monitoring of greenhouse, the correctness of green housing, and drones of effective small cultivation. Well, the following classification is based on numerous applications of effective small cultivation which are of different types.

Figure 9 IoT structure in green housing.

Designed by Freepik (https://www.freepik.com/).

Climate monitoring

Farms are small areas we take as an example drones are useful for forest and horticulture analysis also as they are flying objects and are accessed remotely or by a source program. And lasting the drones are useful in maintaining the log or record for future use by having images, plant health reports, and soil quality check reports. So, having drones can ensure that the farms will have an improved and better production rate than the normal manual structure used before on the farm (Köksal & Tekinerdogan, 2018). To make the greenhouse even smarter the IoT further extended to measure the climate for the plants in the greenhouse by checking the temperature, humidity, the air control system, overheating, and also the light effect and the oxygen level can be maintained (Bhasker & Murali, 2024). This can be done by the implementation of a smart greenhouse system through sensors and making their data on the cloud so that they can share to make some intelligent decisions as mentioned in Fig. 10.

Figure 10 IoT based greenhouse scenario.

Designed by Freepik (https://www.freepik.com/).

Plant monitoring

IoT sensors also deal with the data in images by simply using the cameras. The processing is done by image processing having past data of a plant to recognize the future problem in the plant and having the solution even before the problem even exists so that timely measures are taken that can’t affect the major productivity rate (Windsperger et al., 2019). The past information can be stored on the IoT cloud so it can be saved securely and for long-term use.

Water management

In the greenhouse, water is the key factor to be controlled it is a major component of water so its requirements should be filled properly. Therefore, for this issue, the IoT uses the sensor to solve the problem by giving water by a dripping method so that will compensate for the water level in every plant and every single root gets enough water for growth, not maximum not minimum just a balance quantity and quality to the plant. It also gathers data to normalize the humidity in the soil (Shi et al., 2019).

Solutions of greenhouses by using smartphones

This has been observed since the last few years when electronic devices are integrated with smart-based phones, this came out as an innovation of the world of technology along with smartphones and these are considered the Internet of thing’s drivers. For bringing versatility in the field of cultivation different types of software and hardware have been constructed. A proper but incomplete survey of applications based on smartphone-supporting solutions for the greenhouse has been shown (Muangprathub et al., 2019). In Fig. 11A classified illustration of apps based on a smartphone has been shown for smart farming. Moreover, there are various apps recently presented which are serving compatible functionalities. All apps on the smartphone which are enlarged and clarified in Fig. 11 have been considered in horizontal form each having a short description. The mentioned apps are not finite on a smartphone; globally, there are a lot of apps for e-farming that have been developed by developers. According to popularity, this article discussed a few advertised or displayed apps.

Figure 11 Smartphone applications for greenhouse.

Designed by Freepik (https://www.freepik.com/).

Sensors and IoT devices in green housing

Presently people want everything automated with less manpower with less time. These requirements are fulfilled by using IoT devices that detect the data through sensors as input from the physical environment. Sensing devices are programmed according to the need and perform some required tasks. Some main sensors are listed such as PIR, motion detector, temperature, soil moisturizer, barometric pressure, humidity, PH, and ultraviolet, etc. Table 2 shows applications based on cultivation sensors, IoT associations, and their operations are discussed in detail but cultivation applications for smartphones are discussed in Fig. 11.

Table 2 Sensor and working.

Name of sensors	Working mechanism	
Climate sensor	A device with advance second-generation Bluetooth 5 technology. The range about 200 m wireless, easy installation with smart hang swing mechanism, waterproof, 30 days log-maintained system. It collects data every 10 to 30 min. A total of 2 aa batteries.	
Pulse device by kick-starter	Modulator: smart device which gets sensors online. We use the sensor by the pulse at any time by simply declare some rule.	
Master sensor	All the sensors, collectively form a broad network through connections modules. This master sensor is working as a gateway by gathering all the sensor's data and generation rate, and then giving this data to the network by cellular connectivity (Benyezza et al., 2023).
The master sensor is of two types: direct-cellular connection, internet through master sensor (Oguntosin et al., 2023).
Direct-cellular connection: this type of sensor use sim connectivity to form a gateway (Salama et al., 2023).
Internet through master sensor: the functionality is same as a master direct sensor but the node sensor does not give data directly to the internet, it acts as a module for a master sensor to work its gateway.
Cluster: cluster of node sensors are combined on the backbone of a master sensor. the clustering can also increase the range of one greenhouse or combined two to three farms.	
Leaf wetness sensor	It is based on electrical resistance this sensor measures the wetness of the leaf.	

Iot and greenhouse security

Figure 12A shows the cultivation sections are predicting to observe considerable receiving of the Internet of Things (IoT) and it will flourish up new greenhouse through IoT applications and devices. These IoT applications and devices are estimated to have a huge amount of delicate information which is why we need to secure it by applying multiple protocols. Therefore, any data loss will become a serious privacy threat. Using IoT completely in the greenhouse field then it makes more difficult to observe and point out the particular attributes of security concerns and privacy such as different threat models and requirements of security in the order of greenhouse. Some security issues are described in Fig. 12A .

Figure 12 (A) IoT security issues in greenhouse; (B) intelligent security architecture for green housing.

Security requirements

In the case of an IoT-based greenhouse, all the standard security requirements are the same just like in other scenarios. That’s why we have to follow the confidentiality, authentication, integrity, authorization, self-healing, data freshness, and non-repudiation requirements to achieve a secure greenhouse.

Attack taxonomy

Various kinds of attacks exist in the IoT model, which gives the attacker a wide chance to attack by selecting different methodologies that can be on a physical device because an IoT-based greenhouse contains physical devices that are placed in the field, predictable or unpredictable (Zhang et al., 2017). Host, information disruption, and networks are the three essential elements on which the attacks are categorized. We are also facing some other kinds of attacks in IoT greenhouse systems like host properties attacks and network properties attacks are also stolen and break the data.

Threat model

IoT networks and devices in greenhouses are at the edge due to high attack levels. Threat models contain three scenarios that include cloud networks, expansion of native networks, and services of the cloud. The generation of threads may be occurring internally or externally in a network. If a greenhouse device generates any threat, then it is considered the most dangerous attack (Elijah et al., 2018). Within a network, it is hard to sense the affected device. Cultivated devices and networks can be hacked by any opponent and they can use power devices such as portable computers, mobile, or any IoT devices for disturbing the network.

IoT-based greenhouse security model

Internet of Things (IoT) based greenhouse is recently in an under-developing stage due to which it is hard to detect all possible threats. To resolve all the threats, we have dynamic properties to apply multiple security protocols. Security is checked in a secure framework in which all the implementation is done and it is capable of detecting and preventing all attacks which are shown in Fig. 12B.

Now suppose that a hacker developed a new kind of attack due to which he can easily get all the related sensitive information about the greenhouse. In this case, the designed system follows dynamic rules and algorithms to resolve this issue. This model follows the protection, diagnosis, and reaction system to resolve all the attacks. Whenever the attack is detected, an action command is sent through the detection system, and the reaction system stores it, and then the protection system shares its abnormality for the resistance from attacks. After getting orders from the detection system, attacker activities are resolved by the reaction system which sends a response to both of the systems. In this way, the whole system fixes the attack.

Security challenges

Confidentiality of collaborators authentication and access control are the three fundamental security factors of an IoT-based greenhouse. For that reason, the network should be protected from external attacks at the perception layer and the aggregated data should be secured in the network layer. On the application layer, authorized data should only be accessed by authorized users (Stočes et al., 2016). In the perception layer, physical security is the most ordinary issue which means information and hardware security. In the greenhouse physical security of the deployed network is essential because all the IoT devices are placed in the field due to which a single privacy protocol is not enough. Information leakage is another serious security issue this information may consist of sensitive data including the location of the greenhouse (Yang et al., 2017). To overcome these issues, security measures like encryption of data, blocker tags usage, jamming, tag destruction strategy, and change in tag frequency, etc. can be taken. Some important points related to hardware restrictions should be considered in the implementation of policies in intrusion perceptions, key distribution, encryption algorithm, and routing policies because sensor nodes and RFID tags have major differences between them. IoT devices are continuously flowing data into a gateway and when processed data is stored on cloud infrastructure side by side (Singhai & Sushil, 2023). There are multiple security policies such as cryptographic algorithm data filtering, identity authentication, the flow of data control mechanisms, etc. that is used for sensor nodes. Security threats such as wiretapping, cheating, tampering, and replay attacks also survive there (Sicari et al., 2015).

Smart management systems for greenhouses

Nowadays IoT gives a boom to greenhouse green housing due to its efficiency and opens up the doors to new startups. In this article, a few examples have been provided to determine the position of IoT in the field of greenhouse green housing (Jayaraman et al., 2016). In a farm, a par add-on can be deployed using a 3d crop sensor array in any geographical location to monitor humidity, carbon dioxide, and level temperature (Grownetics, 2019). To control the environment ec-1 controller is used to control the environment by switching devices on and off through programming. The first device that connects the global weather data within the field to observation was invented by Arable and the device is named Arable Mark. It deals with making informed decisions with the unheard-of ground truth authentic device and sends information to the user in real-time. Growlink has the highest processing power that coordinates with multiple sensors and all the deployed devices to provide a smart green housing experience (GreenIQ, 2019). It can be expanded by adding it to the network. Growers have a green specialty that is used to save water bills up to 50%. Grofit transmission range infield is up to 200 m based on Bluetooth. The data log is also maintained through this device which stores maximum measurements of 30 days. It can monitor various green housing factors such as humidity, sunlight, temperature, etc. by the usage of this device (ARABLE, 2019).

Allmeteo designed the weather station of Meteohelix which provides authentic, open, and stable solutions for barometrical maintenance for weather requirements. That’s why we can monitor different environmental factors such as solar radiations, dew points, atmospheric pressure, temperature, humidity, measurement, and sun radiations (METEO, 2019).

Smart cultivation xtreme with waspmote plug and sense is considered to be a node of sensor which supports more accurate and reliable useful knowledge related to weather. The wind is measured by a sensor and the condition of rainfall through the technology of optics. The presence of soil morphology and fertilizers is easily analyzed by this sensor in terms of measuring the level of oxygen, the content of water, and the water potential of the soil (Libelium, 2019). Sky Lora station of weather can communicate to a nearby sensor that is mastered through Lora (Vistro, Rehman & Farooq, 2022). Figure 13 shows the connectivity is found then this is considered to be suitable for such locations. Data that is up to meters around 600 can be sent by this station of weather away to a master sensor having a connection Wi-Fi. The pulse Internet of Things automated sensor has been designed by Pycno and this comes in a package that is self-sustained and powered by the smallest panel of solar (Arshad et al., 2022). Wi-Fi is a sensor along with Lora facilitating the device having a port of multiprotocol placed in the bottom. In the future picnic sensors of soil and pulse automated sensors surely will have an integration to activate devices and interaction in the field. Starter kit of crops–temperature of soil 24/7 is an actual-time sensor for monitoring the temperature of the soil. The sensor has a directly plain cellular-based connection and improved accuracy that provides advanced capabilities of sensing (Cropx, 2019).

Figure 13 Selected IoT greenhouse products.

Designed by Freepik (https://www.freepik.com/).

Key challenges and research directions in green houses and modern cultivation

Various scientists have worked tirelessly on IoT cultivation systems, resolving numerous technological challenges and structural issues by conceptualizing and developing diverse IoT greenhouse solutions. Moreover, as highlighted in the literature, some uncommon open problems and challenges can likely be addressed effectively (Patil et al., 2017). Several difficulties are associated with IoT-based smart greenhouse organizations and applications. This study has identified both known and emerging problems and challenges in IoT-based greenhouse management.

IoT green housing platforms

IoT greenhouse systems are more intricate than other IoT devices and require real-time monitoring capabilities under stringent constraints. This necessitates a specialized computing platform with runtime libraries. A Facility Learning Tactic (FLT) requires an appropriate platform, and these facilities can be leveraged using various APIs (Jawad et al., 2017). Additionally, comprehensive systems and libraries should be developed so that greenhouse developers can make practical use of accessible data, classes, code, and other valuable information.

Networking challenges

Networking challenges are not confined to physical applications but also extend to the network layers. Due to the high cost of cabling, wireless communication is crucial for the development of IoT-based greenhouses. The physical implementation shows that the performance of standard transceivers is affected by the social presence, temperature, and various obstacles within the environment where a remote device needs to communicate (Ojha, Misra & Raghuwanshi, 2015). A comprehensive study on IoT greenhouse networking challenges and issues is required.

Organizational challenges

Numerous challenges have arisen in IoT greenhouse systems. Devices in the perception layer are directly exposed to harsh environmental conditions such as rain, high temperatures, extreme humidity, strong winds, and other potential hazards that can damage electric circuits. End devices operating for extended periods rely on low-battery power sources. Therefore, appropriate programming tools and low-power approaches are required because any program failure may immediately result in battery depletion, which is particularly problematic in large, vulnerable fields (Rehman et al., 2022; Botta et al., 2016).

Technical issues

IoT devices in greenhouse environments are typically deployed over a broad area and may be affected by harsh environmental conditions, potentially causing communication failures and sensor malfunctions. The physical integrity of IoT devices and infrastructure is crucial to safeguard equipment from unauthorized access and severe attacks like adverse weather or theft. Additionally, there are technical challenges related to universal platforms, scalability, cost analysis, Quality of Service (QoS), resource optimization, network interference, mobility, and the deployment of LPWAN technologies in smart greenhouses.

Hardware challenges

Several hardware challenges exist in IoT greenhouse systems. Devices at the transport layer are directly exposed to harsh environmental conditions like rain, high temperatures, strong winds, and other potential risks that may damage electronic circuits. End devices must operate reliably over long periods, relying on low-power battery control resources (Hassan, Park & Han, 2023). Thus, appropriate programming tools and low-power approaches are essential because replacing batteries in the event of a failure is challenging, especially in large-scale vulnerable fields.

Business model

The business model for IoT greenhouse technology remains unclear due to numerous fundamental components, including new optional guidelines and ways to change the structure. Developing a sustainable business model requires addressing novel requirements and aligning with the latest prerequisites.

Conclusion

The integration of IoT technology in greenhouse cultivation offers innovative ways to enhance productivity and optimize resource use. This survey provides a comprehensive overview of state-of-the-art IoT applications in greenhouse management, highlighting the key design principles, platforms, and topologies that underpin the effective implementation of IoT infrastructure. It further explores current advancements in greenhouse applications, devices, and sensors, alongside communication standards and emerging technologies. By addressing the challenges and security requirements of IoT-based greenhouse systems, this study offers valuable insights into enhancing the safety and resilience of such systems. Moreover, this article presents a detailed analysis of critical components in IoT-based greenhouse management, including technologies, enterprise trends, and government policies, aiming to support stakeholders in their adoption of IoT. Governments are increasingly encouraging the implementation of IoT in greenhouse farming, and many major corporations are investing in developing new IoT-based solutions for modern agriculture. This comprehensive review provides useful insights for scientists, experts, agricultural professionals, and policymakers working in the IoT and greenhouse management fields. Our research demonstrates how IoT-based greenhouse systems can improve productivity, resource efficiency, and data-driven decision-making. Governments are increasingly supporting IoT in greenhouse farming, and many corporations are investing in new solutions. This comprehensive review offers crucial insights for scientists, experts, and policymakers, contributing significantly to the growing field of IoT-based greenhouse management.

Future work: In the future, IoT-based greenhouse management should focus on integrating AI and machine learning for predictive analytics and decision-making. Developing universal standards and incorporating blockchain technology will enhance interoperability and data security. Research on low-cost, energy-efficient devices, edge computing, and sustainability metrics will ensure scalable, eco-friendly solutions. Global case studies will help share best practices, advancing innovative and sustainable agriculture.

Additional Information and Declarations

Competing Interests

Author Contributions

Data Availability

The authors declare that they have no competing interests.

Attique ur Rehman conceived and designed the experiments, authored or reviewed drafts of the article, and approved the final draft.

Songfeng Lu performed the experiments, authored or reviewed drafts of the article, and approved the final draft.

Muhammad Awais Ashraf conceived and designed the experiments, prepared figures and/or tables, authored or reviewed drafts of the article, and approved the final draft.

Muhammad Shahid Iqbal performed the experiments, authored or reviewed drafts of the article, and approved the final draft.

Awais khan Nawabi analyzed the data, authored or reviewed drafts of the article, and approved the final draft.

Farhan Amin analyzed the data, performed the computation work, prepared figures and/or tables, authored or reviewed drafts of the article, and approved the final draft.

Rashid Abbasi analyzed the data, authored or reviewed drafts of the article, and approved the final draft.

Isabel de la Torre performed the computation work, authored or reviewed drafts of the article, and approved the final draft.

Santos Gracia Villar performed the computation work, prepared figures and/or tables, authored or reviewed drafts of the article, and approved the final draft.

Luis Alonso Dzul Lopez performed the computation work, authored or reviewed drafts of the article, and approved the final draft.

Md. Belal Bin Heyat conceived and designed the experiments, performed the experiments, analyzed the data, prepared figures and/or tables, authored or reviewed drafts of the article, and approved the final draft.

The following information was supplied regarding data availability:

This is a literature review.

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
