# Peer review of "The role of Internet of Things (IoT) technology in modern cultivation for the implementation of greenhouses"

_PeerJ Computer Science, doi:10.7717/peerj-cs.2309_

## Round 0.1 · original submission · Major Revisions

The methodology and motivation need to be included. Additionally, the authors should provide clear research directions.

Reviewer 1 has suggested that you cite specific references. You are welcome to add it/them if you believe they are relevant. However, you are not required to include these citations, and if you do not include them, this will not influence my decision.

**Language Note:** The review process has identified that the English language must be improved. PeerJ can provide language editing services - please contact us at [email protected] for pricing (be sure to provide your manuscript number and title). Alternatively, you should make your own arrangements to improve the language quality and provide details in your response letter. – PeerJ Staff

Reviewer 1 ·

Basic reporting

no comment

Experimental design

no comment

Validity of the findings

no comment

Additional comments

Suggested Decision: Major Revision
Summary: In this paper, authors reviewed Internet of Things (IoT) technology in modern cultivation. The use of IoT in greenhouses and modern cultivation is very interesting. There is limited literature reported on this and thus topic is interesting and in the scope of the journal. However, there are various serious concerns related to this paper. Based on these concerns, I suggest major revision for this paper.

Comments:
• The problem statement should be briefly described in the Abstract section. The implications, challenges, and future research should be mentioned in the abstract section.
• I have checked that in the introduction section, the IoT abbreviation is not mentioned. Please carefully check this point in your manuscript. In addition, the authors did not discuss the summary of the most recent state of the art in his area. I suggest to consult the following related articles for further clarification on the idea.

“Crop Yield Prediction Using Machine Learning Approaches on a Wide Spectrum” in Computers, Materials & Continua, vol. 72, No. 3, pp. 5663-5679, 2022. DOI: 10.32604/cmc2022.027178.

“Implementation of a LoRaWAN Based Smart Agriculture Decision Support System for Optimum Crop Yield” Sustainability 2022, Vol. 14, Issue 2, 827. DOI: 10.3390/su14020827
• Therefore, they should briefly discuss the state of the art and the summary and also the difference between this work and others.
• The problem statement and benefits of this research should be highlighted in the introduction section.
• The contribution is not clear and therefore it should be more polished and should be the part of introduction section.
• The research gap should be indicated. The aim of the article should also be provided in the Introduction (not just in the Abstract).
• In sections 2 and 3, there is a need to add the most recent state-of-the-art in this area including:

“GSM based Solar Automatic Irrigation System using Moisture, Temperature and Humidity Sensors” 5th International Conference on Engineering Technology & Technopreneurship (ICE2T), ISBN 978-1-4799-4621-1, Kuala Lumpur, Malaysia, 18-20 September 2017. DOI: 10.1109/ICE2T.2017.8215945
“Deployment of Wireless Sensor Network and IoT platform to Implement an Intelligent Animal Monitoring System” Sustainability 2022, Vol. 14, Issue 10, 6249. DOI: 10.3390/su14106249

• In section 4, there is a need for the most recent research studies along with references. Currently, there is no reference. Authors are suggested to please study the following article “Intelligent greenhouse monitoring and control scheme: An arrangement of Sensors, Raspberry Pi based Embedded System and IoT platform.” Indian journal of science and technology 13 (2020): 2811-2822.

• In section 5, the title is not clear. Please modify the revised version.
• Section, 8 is about the challenges in this area. Please modify the title. Currently, It is not clear.
• The conclusion section should be more précised. Currently, it is very long. Future work should be part of this section.

Reviewer 2 ·

Basic reporting

In this paper, the author performs a review of Internet of Things technology in Modern Cultivation for the Implementation of Greenhouses.

Experimental design

The paper presentation needs special attention. Please revise your paper according to the below suggestions.
1) The abstract and the conclusion section should be more polished. The future work should be the part of conclusion section.
2) The paper organization missing at the end of the paper. Please mention this at the end of the introduction section.
3) The contribution should be mentioned in bullets.
4) Section 2 state of the art of related work is missing. Please add this section, and cite the most recent state of the art in this area. In addition, the difference between state-of-the-art and your research should be mentioned in this section.

Validity of the findings

5) Please reduce section 3 and remove unnecessary subsections, i.e. 2.10.1 Networks and Technologies. It is advised to add the most recent studies in other sections.
6) The section is shortened, please increase by adding more studies in this section.

Additional comments

7) More polish challenges section.
8) Check English and grammar for typos and mistakes.

·

Basic reporting

No Comment

Experimental design

No Comment

Validity of the findings

No Comment

Additional comments

I have gone though this study and found that it needs serious attention. Thus, it should be improved in terms of the following points.

1- There is unnecessary information in the abstract, thus, it should be polished. We expect a scientific manuscript presenting the results of the study, not a report summarizing a study. I think we should see the following information in a few sentences in the abstract.
1.1. What was done in this study?
1.2. Why was this study done?
1.3. What are, the implications of this study?
1.4 The motivation and the contribution should be mention in the introduction section
The abstract section should answer the above questions and should be more polished.

2- In the conclusion section, a summary is required. The implication of your study and the success achieved should be summarized in just a few sentences.
3- The paper presentation, organization, and structure of the article should be revised to make the subject more understandable.
4- Most references on this subject should be added. The number of references is not small. However, in this type of study, I think there should be much more up-to-date references.
5- I think that the Result, Discussion, and Conclusion sections should exist separately for such an article.
6- Authors are advised to add the latest citations, please read and cite.
7- Overall, there are still some minor parts that the authors did not explain clearly. There are typos and grammar mistakes and as a result, I am going to suggest a Major revision of the paper in its present form.

---

## Round 0.2 · accepted · Accept

Paper is now accepted. The authors improved paper in revision

Reviewer 1 ·

Basic reporting

The article is well written.
Literature references and sufficient background have been provided.
Figures and Tables are of good quality.

Experimental design

Article content is within the Aims and Scope of the journal and article type.
Methods are described with sufficient detail.

Validity of the findings

conclusion is well stateed.

Additional comments

Thanks for providing the revised paper. In the revised version, the authors improved the quality of figures and also revised the paper in different sections. In addition, I have checked and found that improvements are satisfied and therefore I recommend it for publication. Thanks.

Reviewer 2 ·

Basic reporting

Overall changes are satisfied. The English is improved in the revised version.

Experimental design

The authors improved the paper by adding contributions, and implications of research. The presentation of the overall paper is improved.

Validity of the findings

Now the paper is in a better state than the previous version. The newly added challenges are more interesting and have an interest of the readers.

·

Basic reporting

I am pleased to see that authors revised paper according to my suggestions. However, There are grammar , typos and English were remains as minor. So, Please take a look and improve in the final version. Good luck

Experimental design

No comment

Validity of the findings

No Comment

Additional comments

No Comment